# Integration of Phosphoproteomics and Transcriptome Studies Reveals ABA Signaling Pathways Regulate UV-B Tolerance in *Rhododendron chrysanthum* Leaves

**DOI:** 10.3390/genes14061153

**Published:** 2023-05-25

**Authors:** Qi Sun, Xiangru Zhou, Liping Yang, Hongwei Xu, Xiaofu Zhou

**Affiliations:** Jilin Provincial Key Laboratory of Plant Resource Science and Green Production, Jilin Normal University, Siping 136000, Chinayangliping781124@163.com (L.Y.); xuhongwei@jlnu.edu.cn (H.X.)

**Keywords:** UV-B stress, ABA, stomatal, *Rhododendron chrysanthum*, phosphorylated proteomics, transcriptome

## Abstract

The influence of UV-B stress on the growth, development, and metabolism of alpine plants, such as the damage to DNA macromolecules, the decline in photosynthetic rate, and changes in growth, development, and morphology cannot be ignored. As an endogenous signal molecule, ABA demonstrates a wide range of responses to UV-B radiation, low temperature, drought, and other stresses. The typical effect of ABA on leaves is to reduce the loss of transpiration by closing the stomata, which helps plants resist abiotic and biological stress. The Changbai Mountains have a harsh environment, with low temperatures and thin air, so *Rhododendron chrysanthum* (*R. chrysanthum*) seedlings growing in the Changbai Mountains can be an important research object. In this study, a combination of physiological, phosphorylated proteomic, and transcriptomic approaches was used to investigate the molecular mechanisms by which abiotic stress leads to the phosphorylation of proteins in the ABA signaling pathway, and thereby mitigates UV-B radiation to *R. chrysanthum*. The experimental results show that a total of 12,289 differentially expressed genes and 109 differentially phosphorylated proteins were detected after UV-B stress in *R. chrysanthum*, mainly concentrated in plant hormone signaling pathways. Plants were treated with ABA prior to exposure to UV-B stress, and the results showed that ABA mitigated stomatal changes in plants, thus confirming the key role of endogenous ABA in plant adaptation to UV-B. We present a model that suggests a multifaceted *R. chrysanthum* response to UV-B stress, providing a theoretical basis for further elaboration of the mechanism of ABA signal transduction regulating stomata to resist UV-B radiation.

## 1. Introduction

*R. chrysanthum* is a perennial dicotyledonous plant of the Rhododendron family. This plant is an endemic and threatened species of the Chinese Changbai Mountains, which have a harsh environment with thin air and intense solar radiation, from which UV-B radiation is the main abiotic stressor in the region [1]. UV-B can induce DNA damage that can lead to mutation or apoptosis; it also causes oxidative stress in plants, which reduces cell integrity and alters plant morphology and physiology [2]. Over a long period of adaptive evolution, *R. chrysanthum* has evolved complex mechanisms to cope with UV-B stress and protect itself from harm, making it an important resource for the study of plant resilience [3].

In previous studies, the physiological changes of chlorophyll content and antioxidant enzyme activities induced by abiotic stress were extensively studied, and the responses of carbohydrates and amino acids in *R. chrysanthum* to UV-B stress were comprehensively analyzed by transcriptional and metabolic analysis, in order to reveal its adaptation mechanism to abiotic stress [4,5]. However, there are few reports about ABA regulating *R. chrysanthum* resistance to UV-B irradiation. Among different hormones, SA, MJ, and ethylene (ET) contribute to resistance to biotic stresses, while ABA is involved in several abiotic and biotic stress conditions, and is therefore considered to be an important multifunctional compound. Under abiotic stresses, the accumulation of ABA causes the stomata to close in order to conserve water. Stomatal closure helps to maintain the dynamic balance needed for normal growth, including photosynthetic carbon sequestration and transpiration, while upregulated genes promote osmoregulation in the leaves [6,7,8]. The increase in ABA in plants is beneficial to abiotic stress adaption. Pyrabactin Resistance 1 (PYR)/PYR1-like (PYL) is known to be a receptor for abscisic acid (ABA), and the combination of PYLs and ABA prevents PP2C from inhibiting SnRK2 and activating complex signal transduction pathways downstream of ABA [1,9,10]. SnRK2 kinase may be involved in the ABA signaling pathway that induces stomatal closure [11,12]. The stomata are the earliest and fastest organs to respond to ABA, and stomatal traits are among the crucial indicators of how well plants can adapt to their ecological surroundings. The decrease in stomatal opening can reduce the damage caused by stress and activate the defense system of plants, which are usually interrelated. These processes control the transpiration rate, leading to the reduction in transpiration and water loss, and thus enhancing the adaptation of plants to abiotic stress [13,14].

This experiment is the first to combine transcriptomics and phosphorylated proteomics with physiological analysis in order to identify genes in the plant hormone signaling pathway in response to UV-B radiation using RNA-seq techniques. Phosphorylated proteins in the *R. chrysanthum* signaling pathway were explored from phosphorylated levels, which were specifically altered under UV-B radiation. Furthermore, protein phosphorylation is an indispensable post-translational modification for stress signaling and the control of various biological processes in plants [15]. Protein phosphorylation can provide additional insights into the mechanisms of plant responses to abiotic stresses, and it is therefore essential to study the characterization of protein phosphorylation modifications in a variety of biotic and abiotic stress domains in order to investigate protein function [16]. In previous studies, changes in the response to cold stress and its mechanisms were revealed from the phosphoproteome in *R. chrysanthum* leaves [17]. Finally, the results showed that external application of ABA could effectively alleviate the negative effects of abiotic stress on plants by adjusting the morphological characteristics and spatial distribution pattern of their stomata [18].

In this study, we explored phosphorylated proteins in the *R. chrysanthum* signaling pathway that are specifically altered in response to UV-B stress at the transcriptome and phosphorylated proteome levels. These results provide a basis for us to better understand the response mechanism of *R. chrysanthum* to UV-B radiation.

## 2. Materials and Methods

### 2.1. Plant Materials and Treatment

Experimental materials were selected from *R. chrysanthum* histoculture seedlings grown in the Changbai Mountains at an altitude of approximately 1300–2650 m, and then transported to the laboratory and kept in the artificial climate chamber that simulates an alpine environment [19]. The relative humidity of the artificial climate chamber was 60%; the temperature ranged from 16 °C to 18 °C; light conditions were maintained for 14 h periods and the dark conditions for 10 h periods; white fluorescent lamps were used at 50 μmol (photons) m^−2^ s^−1^. The 8-month-old seedlings were divided into two groups: the experimental group received an added 8 h of UV-B exposure per day; the control group received normal light for plants. Measurements were taken 2 days after UV-B stress was applied, with three biological replicates for each group. The artificial radiation of UV-B (295–315 nm) and PAR (400–700 nm) were used in this study to treat the *R. chrysanthum* samples [20,21]. Using long-pass filters, UV intensity meters (Sentry Optronics Corp., ST-513, Taipei, China), and transfer functions of photometers (TES Electric Electronics, Taipei, China, TES-1339Light Meter Pro), the irradiance effectively received by the samples for UV-B and PAR were 2.3 Wm^−2^ UV-B and 50 μmol (photons) m^−2^ s^−1^ par (China).

The control group was transplanted into 1/4 MS medium without ABA, and the experimental group was transplanted into 1/4 MS medium with ABA (100 umol/L). After transplantation, the *R. chrysanthum* samples were cultured in an artificial climate chamber for 6 days, and then exposed to UV-B radiation (2.3 Wm^−2^) for 8 h/d, and the indicators were determined after UV-B radiation treatment for 2 days.

### 2.2. Determination of Stomatal Characteristics

The stomata were collected from the middle of the leaves of the *R. chrysanthum* samples in 2 mm × 2 mm pieces against the leaf veins, and the stomata were collected by applying transparent nail polish evenly on the abaxial surface of the leaf and producing clinical films [22]. Three clinical films were produced for each treatment, five fields of view were randomly selected in each clinical film, and a microscope camera system (Model C-SHG 1, Nikon Corp, Toyko, Japan) was used to observe and take pictures. Fifteen pictures of stomatal structure were obtained for each treatment, and three pictures were randomly selected. Thereafter, the length, width, and area of stomata were measured by ImageJ software [23].

### 2.3. RNA-Seq Library Construction and Sequencing

The mRNA of the polyA tail was enriched using magnetic beads with OligodT. rRNA removal occurred as follows: rRNA was hybridized with the DNA probe, RNAseH selectively digested the DNA/RNA hybrid strand, and DNaseI digested the DNA probe and purified the desired RNA. A proper amount of interrupting reagent was added to the obtained mRNA in order to fragment it under high temperature conditions, and a strand of cDNA was synthesized using the interrupted mRNA as the template, and then a two-strand synthesis reaction system was configured to synthesize a two-strand cDNA; subsequently, a kit was used to purify and recover, sticky end repair, base “A” was added to the 3 ‘end of cDNA, a joint was connected, and then the fragment size was selected. Finally, PCR amplification was performed. The constructed library was inspected by Agilent2100Bioanalyzer and Abisteponeplus Real-Time Time CrAnalyzer, and sequenced after being qualified.

Six cDNA libraries of *R. chrysanthum* were constructed using Shenzhen Huada illumina platform for UV-B stress and normal light growth. The raw reads of the six samples ranged from 48.99 to 50.62 MB, and the base of low Quality < 20 was lower, indicating good sequencing quality. After data filtering, clean reads ranged from 42.11 MB to 43.29 MB. The effective reading sequence Q20s of the two groups were 98.29% to 98.44% and 98.2% to 98.43%, respectively. Q30s were 94.83% to 95.22% and 94.58% to 95.16%, respectively. The results showed that the sequencing results were of good quality and the data could be used for further assembly [4].

### 2.4. De Novo Assembly and Sequence Annotation

In this experiment, SOAPnuke, a filtering software developed by BGI, was used for statistics, and Trimmomatic was used for filtering. The experimental reference genome version is Rhododendron_lapponicum. Since there is no reference genome sequence of *R. chrysanthum*, the high-quality data (clean reads) were reassembled by Trinity software. The total reads of the two groups were 103,951 and 100,725 respectively, the total lengths were 105,212,398 and 104,074,971 bp, respectively, and the average lengths of 1010 and 1018 bp N50 were 1605 and 1598 bp, respectively. The results show that the assembly quality of the sequencing results is good and can be used for further analysis and comparison.

BLASTX was utilized in order to search in NCBI public databases such as Nr, NT, Pfam, KOG, and Sissspro. In order to annotate gene ontology terms, we used the best shot with Nr annotation submitted to the BLASTX Blast2GO program. Analysis of transcript metabolic pathways was performed using the KEGG database [24].

### 2.5. Differential Expression Analysis

In this paper, the effective reading sequence (clean reads) was compared to the genomic sequence using Bowtie2 software, and then the gene expression level of each sample was calculated by RSEM. The gene expression of each transcript was analyzed by FPKM method. FC represents the differential factor of gene expression. FC > 2 or <1/2 and significance *p* < 0.001 were used as the thresholds by which to evaluate gene differential expression.

### 2.6. Protein Extraction and Digestion

Two grams of sample were added to liquid nitrogen for grinding. After adding 4 L of lysis buffer (8 M urae, 1% Triton-100, 10 mM dithiothreitol, and 1% Protease Inhibitor Cocktail), ice was used to lyse the samples three times using a high-intensity ultrasonic machine (Scientz, Shanghai, China). We separated the residual debris via centrifugation for 10 min at 4 °C at 20,000× *g*. A 20% TCA solution was used to precipitate proteins at −20 °C for 2 h, followed by centrifugation at 12,000× *g* for 10 min at 4 °C and discarding the supernatant. The remaining precipitated material was washed three times with 100 mL 90% cold acetone, centrifuged for five minutes at 4 °C at 12,000× *g*, and collected to airdry. Proteins were resolubilized with 8 M urea and protein concentrations were determined using a 2-D Quant kit (GEHealthcare, Chicago, IL, USA). In order to improve accuracy, each group was repeated three times.

Afterward, the protein solution was mixed with dithiothreitol and reduced at 56 °C for 30 min. Thereafter, iodoacetamide was added and incubated in the dark for 15 min at room temperature. Trypsin was added at a mass ratio of 1:50 (trypsin: protein), the samples were digested overnight at 37 °C, and the released peptides were collected via centrifugation [17].

### 2.7. Phosphopeptide Enrichment

The peptide mixture was first incubated by shaking with IMAC microspheres loaded in a suspension containing buffer (50% acetonitrile/6% trifluoroacetic acid). The phosphopeptide-rich IMAC microspheres are collected by centrifugation and the supernatant is removed. In order to remove non-specific adsorbate polypeptides, the IMAC microscope can be progressively treated with 50% acetonitrile /6% trifluoroacetic acid and 30% acetonitrile /0.1% trifluoroacetic acid. A solution containing 10% NH_4_OH was added in order to elute the enhanced phosphopeptides from the IMAC microspheres. Finally, the lyophilized supernatant containing phosphopeptides was collected [25].

### 2.8. LC-MS/MS Analysis and Database Search

The peptides were dissolved in mobile phase A, which contained 0.1% (*v*/*v*) formic acid aqueous solution, for liquid chromatography, and were then separated using an EASY-nLC1000 ultra-high-performance liquid phase system. The gradient of the liquid phase is 500 nL/min, 0–40 min, 4–22% B; 40–52 min, 22–35% B; 52–56 min, 35–80% B; and 56–60 min, 80% B. A UHPLC system was used to separate peptides, introduce them into the NSI ion source for ionization, and then introduce them into the Q-accurate mass spectrometer for analysis. When the scanning range of secondary mass spectrometry was set to a fixed starting point of 100 *m*/*z*, and the secondary scan resolution to 35,000, the scanning range of primary mass spectrometry was set to 350–1800 *m*/*z* and the resolution to 70,000. Thirdly, in the preparation of secondary mass spectrometry analysis, 31% fragmentation energy was used in order to fragment the collision pool. Automatic gain control (AGC) was set to 1E5, the maximum injection time was set to 100 ms, and the dynamic exclusion time for the tandem mass spectrometry scan was set to 30 s [26].

Data from raw mass spectrometry were analyzed using the MaxQuant program (version 1.5.2.8) [27]. Search parameter settings: the database was Rhododendron_ laponicum _ 313330 (45,945 sequences), the false positive rate (FDR) of random matches was calculated by adding the reverse library, and the effect on contaminating proteins was removed; the digestion mode was trypsin/P. For the first search, the mass error tolerance of the primary parent ion was set to 20 ppm, the primary search was set to 5 ppm, and 0.02 Da for the secondary fragment ion. The alkylation of cysteine was a fixed modification, and the oxidation of methionine, acetylation of N-terminal amines, deamidation (NQ), and phosphorylation of threonine, serine, and tyrosine were variable modifications. During protein identification and PSM identification, TMT-6plex was used as the quantification method, and 1% FDR was used as FDR.

### 2.9. Bioinformatics

The GO annotation of proteins is classified into three major categories, namely: biological processes, cellular components, and molecular functions. The context of the identified proteins was examined using Fisher’s exact test, and GO enrichment tests with *p*-values < 0.05 were considered significant.

The Kyoto Encyclopedia of Genes and Genomes (KEGG) database was used for the enrichment analysis of pathways. Fisher’s exact double-end test was used to test for differential proteins and pathway enrichment tests with *p* values < 0.05 considered significant. Finally, the pathways were classified according to the KEGG website pathway hierarchy classification method.

Three-dimensional homology structure models of phosphoproteins were built using the SWISS-MODEL comparative protein modeling server8, and the structures were commercialized and optimized using Swiss-Pdb Viewer software (version 3.7) [28].

## 3. Results

### 3.1. Transcriptomic Analysis of R. chrysanthum under UV-B Stress

Transcriptome analysis of *R. chrysanthum* subjected to control (CK) and UV-B (NA) conditions revealed that 12,289 differentially expressed genes with FC >2 or <1/2 and significance *p* < 0.001, including 7804 upregulated and 5634 downregulated genes, were identified (Figure 1A). The biological samples treated with PAR or UV-B were clearly separated; nevertheless, biological replications under the same conditions appear to be very similar based on principal component analysis (PCA) (Figure 1B). The enrichment factor, q-value, and number of genes enriched in the pathway were used to calculate KEGG enrichment. The top 16 KEGG pathways were selected for display (Figure 1C). The most significant enrichment of DEGs by KEGG in this study was observed in ABC transporters, Glycosaminoglycan degradation, and plant hormone signal transduction. This differentially expressed set of functional genes may play a significant role in *R. chrysanthum’s* UV-B radiation response. Since this study focused on the abscisic acid metabolic pathway, the phytohormone signaling pathway was used as the entry point for the study. A number of genes related to phytohormone signaling were analyzed (Figure 2A), including ABA, IAA, JA, SA, Cytokinine, GA, Ethylene, and BA. As far as genes involved in the ABA signaling pathway are concerned, we noticed the following: UV-B radiation reduced the expression of *PYR/PYL* receptors, as well as *PP2C*, *SnRK2*, and *ABF* genes (Figure 2B).

### 3.2. Phosphorylated Proteomics Analysis of R. chrysanthum under UV-B Stress

We identified 2872 phosphopeptides of 2508 phosphoproteins separately in *R. chrysanthum*. Of these, 128 phosphorylation sites (*p* < 0.05, fold change > 1.2) were identified on 109 differentially phosphorylated proteins (Figure 3A). Phosphorylation levels increased at 42 sites and decreased at 86 sites after UV-B stress. Among the 128 phosphorylation sites, 103 are phosphorylated on Serine (Ser) residues, accounting for 80% of all sites, and 25 are phosphorylated on threonine (Thr) residues, accounting for 20% of all sites (Figure 3B; Appendix A).

The GO classification statistics of differentially phosphorylated proteins in *R. chrysanthum* indicated that the GO entries for differentially phosphorylated proteins were significantly enriched in three major functional categories (Figure 3C). In the category of biological processes, there were six functional subcategories, and the numbers of differentially phosphorylated proteins in the two functional subcategories of cellular processes and metabolic processes were higher; in the category of cellular components, there were four functional subcategories, and the numbers of differentially phosphorylated proteins in the two functional subcategories of cells and membranes were higher; in the category of molecular functions, there were five functional subcategories, and the numbers of differentially phosphorylated proteins in the two functional subcategories of binding and catalytic activity were higher.

### 3.3. Three-Dimensional Structure Modeling of UV-B Stress-Responsive Phosphoproteins

In order to gain a clearer understanding of the phosphoproteins that undergo functional changes in ABA signaling in *R. chrysanthum* under UV-B irradiation, a simulation of the molecular structure of phosphorylated proteins was conducted. Two statistically acceptable homology models, with phosphorylation sites located within the 3D structural model, were constructed using the Swiss model (Figure 4). Under UV-B stress, ABF2 protein was phosphorylated at Thr422, SnRK2 was phosphorylated at Ser154, and both phosphorylation levels were elevated.

### 3.4. Transcriptomic and Phosphorylated Proteomic Interaction Analysis to Explore the Response of R. chrysanthum to UV-B Stress

In order to clearly understand the relationship between the ABA signaling pathway genes of *R. chrysanthum* under UV-B radiation and phosphorylated modified proteins, we combined them to build a network to more intuitively understand the correlation between the two. As can be observed from Figure 5, the expression levels of *PYR/PYL*, *ABF*, and *PP2C* are low, while the expression levels of *SnRK2* are high. Both SnRK2 and ABF proteins were upregulated during phosphorylation.

### 3.5. Effect of Exogenous ABA on Stomata of R. chrysanthum after UV-B Stress

In order to determine whether ABA plays a role in the response of *R. chrysanthum* to UV-B stress, a validation experiment was conducted in which temporary patches of *R. chrysanthum* pores were prepared and observed under a microscopic imaging system, and photographs were taken for measurement and data analysis (Figure 6). Compared with the PAR group, UV-B stress significantly reduced stomatal width and area. Compared with the UV-B group, the area and width of stomata in the ABA + UV-B group were significantly increased. When *R. chrysanthum* was subjected to UV-B stress, the stomatal width area was reduced, thus reducing transpiration and water loss.

## 4. Discussion

This study was designed to uncover the primary response of *R. chrysanthum* to the stress of UV-B, a major limiting factor for plant growth at high latitudes and altitudes. A large number of studies have shown that UV-B enhancement can affect physiological and biochemical processes in many plant species, ultimately leading to changes in crop morphology. In our work, through transcriptomic and phosphorylation proteomics analysis of the response to UV-B stress in *R. chrysanthum*, we found that ABA signaling in plants can regulate the closure of the stomata by associated phosphorylated proteins, which would partially counteract the damage caused by UV-B radiation; thus, we were able to better investigate the mechanisms of plant response to UV-B stress.

Abscisic acid (ABA), a phytohormone, has multiple functions in plant regulation for the development around and tolerance of various biotic and abiotic stresses [18]. When plants are subjected to adverse environmental stresses, such as drought, low temperature, and salinity, ABA levels increase, activating ABA signaling, and being sensed by defense cells, promoting stomatal closure and inhibiting stomatal opening. In particular, ABA-dependent and ABA-independent signaling pathways, which are also intertwined, mediate the responses of plants to abiotic stresses. Our experimental data also proved that the ABA-dependent transduction pathway in *R. chrysanthum* changed after UV-B stress [29]. Transcriptome was used to analyze the functional annotation of transcripts in plant leaf tissues in this paper. Under UV-B stress, the enrichment pathway is mainly plant hormone signal transduction. In the previous study, plants were found to have a similar pathway in response to abiotic stress [30,31]. As integrators of environmental signals, hormones play an important role in regulating plant stress responses. An earlier study found that ABA synthesis and metabolic genes are suppressed by abiotic stress in plants, suggesting that UV-protective ABA regulatory mechanisms are also present in *R. chrysanthum* [24]. The ABA signaling pathway consists of four core components that, together, form a dual negative regulatory system that regulates the response to ABA. A co-receptor complex formed by PYR/PYL and PP2C senses ABA. The perception of ABA causes PP2C phosphatase activity to be blocked, and the kinase SnRK2 associated with it is released from PP2C inhibition [32]. It is possible that the net reduction in ABA is due to the decreased expression in corresponding receptors and in the gene encoding *SnRK2*, which is relatively upregulated under UV-B conditions [33]. The activated SnRK2 can act on downstream transcription factors (ABF) to regulate the expression of genes involved in ABA signaling. The N-terminal of the AREB/ABF transcription factor contains three conserved domains (C1, C2, C3). The C-terminal conserved domain (C4) is located at the C-terminal of the protein and contains a highly conserved alkaline leucine zipped structure that binds DNA and other proteins, and requires phosphorylation to be activated [34,35]. The results showed that ABA activated SnRK2 protein kinase and phosphorylated Thr residues in the T site of ABF conserved region. SnRK2 protein kinase regulates ABA signaling through phosphorylation of conserved ABF sites. Thus, ABA signaling is involved in the UV-B signaling response in *R. chrysanthum*.

Phosphorylation is the most prevalent post-transcriptional modification, and phosphorylation modifications of key proteins play an efficient role in cellular signaling during signal perception and transduction in response to abiotic stresses in plants [36]. Therefore, studying the characteristics of protein phosphorylation modifications in a variety of biotic and abiotic stress domains is essential in order to investigate protein function. Although it has been shown that protein kinases and protein phosphorylation processes are involved in stomatal movement, the specific mechanisms of action are only just beginning to be investigated. Using phosphorylation proteomics, we analyzed the key factors involved in UV-B resistance in *R. chrysanthum*, explored at the level of protein post-translational modifications, thus providing a bioinformatic basis for further in-depth study of the plant UV-B resistance mechanism. Through bioinformatics analysis, the functional classification of the functional groups in which the phosphorylated differential proteins of *R. chrysanthum* were located after UV-B radiation, when biological process analysis showed greater cellular processes and metabolic processes; cellular component analysis showed more cell and membrane processes; molecular function analysis showed more binding, catalytic activity, and transport activity. Therefore, under UV-B radiation, phosphorylation contributes to the organization of cell components and the modification of proteins. A proteinomic analysis of phosphorylation under abiotic stress shows that the changes of phosphorylation mainly occur in regulatory proteins of signal transduction, transcription, translation, and transportation [37,38,39]. Additionally, in this experiment, a protein kinase (SnRK2) related to ABA signaling was identified. Protein kinases are intracellular information-dependent enzymes that assist in signal transduction and mediate protein phosphorylation in the organism. Protein kinase/phosphatase-mediated protein phosphorylation/dephosphorylation is essential for abscisic acid signaling [40]. Furthermore, we found that the phosphorylation level of ABF, a transcription factor downstream of the *R. chrysanthum* abscisic acid receptor, was significantly upregulated during resistance to UV-B radiation. ABF transcription factor is a type of basic leucine zipper (bZIP) protein, belonging to subgroup A of bZIP transcription factors, which exists specifically in plants and plays a key role in plants’ response to hormones and influences the ability to resist stress [18].

Stomata are small holes surrounded by guard cells on the surface of plant leaves. As an important gas exchange channel between leaves and air, its opening and closing movement control has obvious influence on important biological processes, such as transpiration and photosynthesis [41]. When plants are subjected to abiotic stress, they will adjust their stomatal characteristics to resist damage. When *R. chrysanthum* suffers from UV-B stress, the stomatal width, area, and other indicators are significantly reduced, and the stomatal opening is reduced, thus weakening transpiration and reducing water loss. UV-B causes excessive photoinhibition, so that excessive heat cannot be emitted to the external environment through transpiration. The application of ABA can increase the transpiration rate by increasing stomatal width and area, so that the heat that otherwise cannot be dissipated due to UV-B can be dissipated, and the photosynthetic rate can be increased. Moreover, CO_2_ can easily diffuse into the leaves of *R. chrysanthum* through the stomata, which affects the maximum reaction rate of photosynthesis, thus improving the tolerance of *R. chrysanthum* to UV-B stress.

## 5. Conclusions

For the first time, a comprehensive analysis of the resistance of *R. chrysanthum* to UV-B radiation was conducted by combining phosphorylated proteomics and transcriptomics. When the receptors PYR/PYLs sense the presence of ABA, they bind to each other and further bind to PP2Cs to form a ternary complex, which inhibits the enzymatic activity of PP2Cs and dissociates the PP2Cs–SnRK2s complex, and SnRK2s undergoes autophosphorylation, which subsequently activates downstream substrates, such as transcription factors, through phosphorylation in order to induce stomatal closure and alleviate the damage caused by UV-B radiation in *R. chrysanthum*. The damage of UV-B radiation to *R. chrysanthum* was alleviated, which laid a foundation for the further study of the molecular mechanism of ABA affecting plant resistance to UV-B radiation.

## Figures and Tables

**Figure 1 genes-14-01153-f001:**
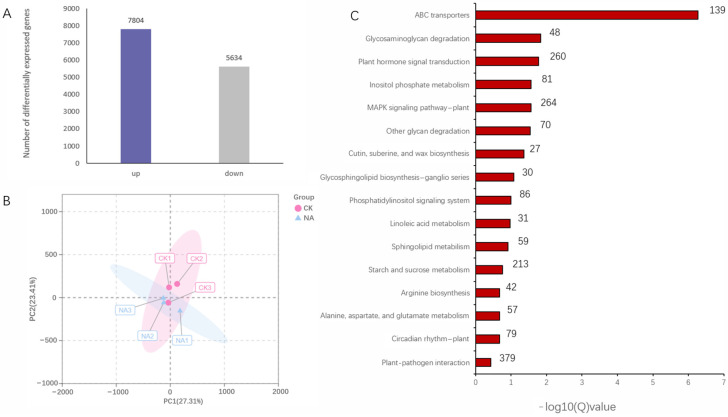
Transcriptome analysis of *R. chrysanthum* under UV-B exposure. (**A**) The number of DEGS in the transcriptome; (**B**) PCA of differential genes; (**C**) KEGG enrichment of differential genes (The number on the edge of each bar represents the number of DEGs).

**Figure 2 genes-14-01153-f002:**
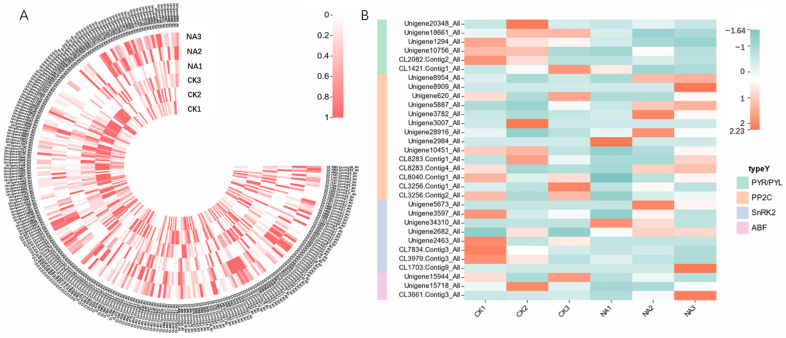
Response of the phytohormone signaling pathway to UV-B radiation in *R. chrysanthum*. (**A**) Heatmaps depict transcript expression profiles associated with phytohormone signaling pathways under UV-B irradiation. A pink box indicates a gene that is upregulated, while a white box indicates a gene that is downregulated; (**B**) ABA signaling pathway transcript expression profile. An orange box represents genes that have been upregulated, while a green box represents genes that have been downregulated. (The value is the log2 fold change (log2(FC)) of each gene.).

**Figure 3 genes-14-01153-f003:**
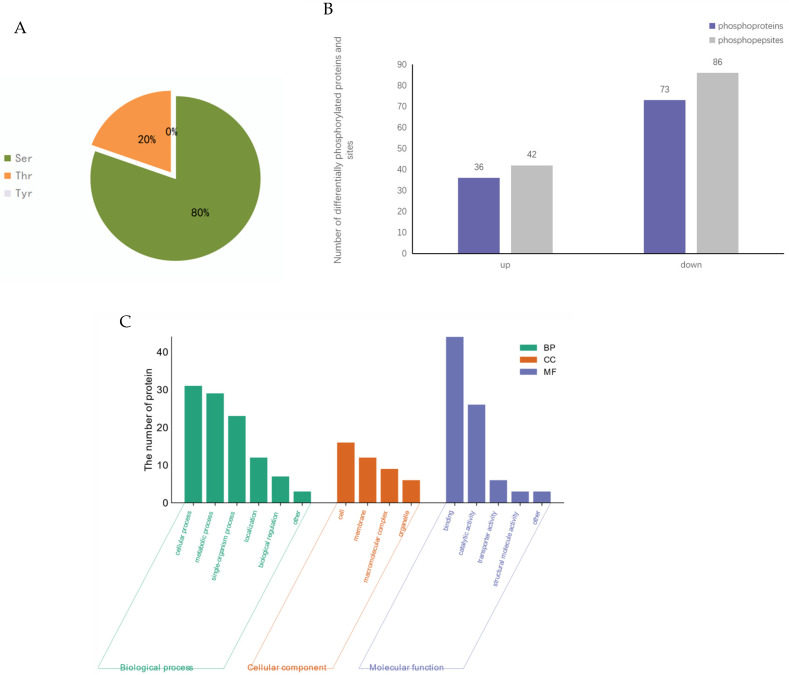
Phosphorylation proteome analysis of *R. chrysanthum* under UV-B irradiation. (**A**) Classification of differentially phosphorylated sites; (**B**) Number of differentially phosphorylated proteins and sites; (**C**) GO classification of differentially phosphorylated protein.

**Figure 4 genes-14-01153-f004:**
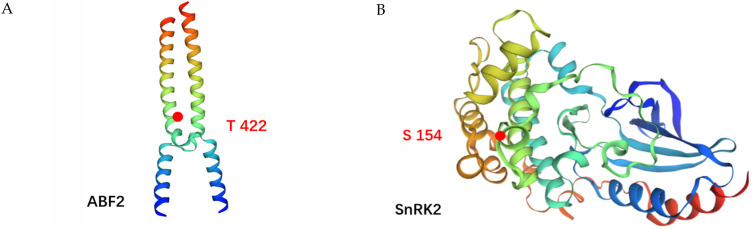
Three-dimensional homologous model of phosphorus protein in response to UV-B irradiation related to ABA signaling of *R. chrysanthum*. (**A**) ABF2; (**B**) SnRK2. The areas with increased phosphorylation levels are marked with red circles.

**Figure 5 genes-14-01153-f005:**
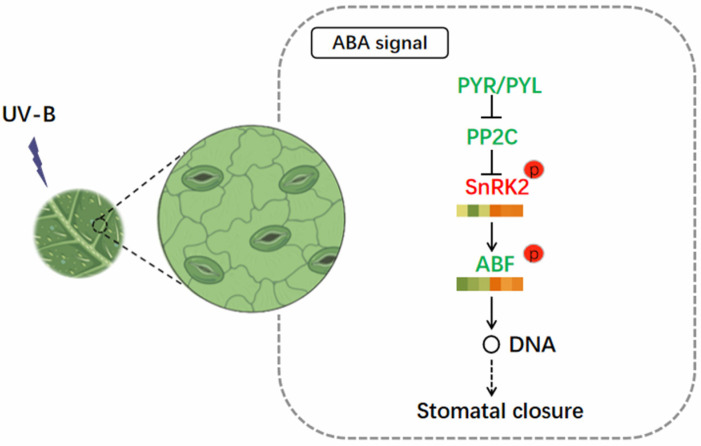
Study on phosphorylated protein and gene levels in ABA signal transduction pathway of *R. chrysanthum* irradiated by UV-B. DEGs are filtered using log2 FPKM (transcript fragments per kilobase per million mapped reads). Genes that are downregulated or upregulated are indicated by green and red text, respectively. The green and yellow boxes represent phosphorylated downregulated and upregulated proteins, respectively.

**Figure 6 genes-14-01153-f006:**
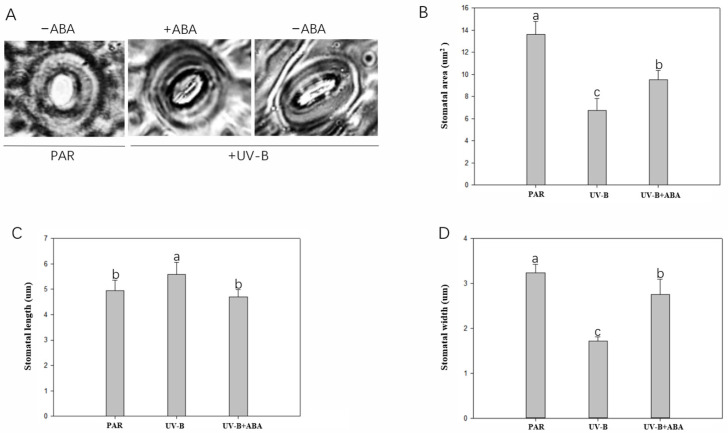
Effects of different treatments on stomata of *R. chrysanthum*. (**A**) Stomatal morphology changes under different treatments; (**B**) Stomatal area; (**C**) Stomatal length; (**D**) Stomatal width.

## Data Availability

The data used in this study are available from the corresponding author on submission of a reasonable request.

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
