# Peer review of "Integration of Phosphoproteomics and Transcriptome Studies Reveals ABA Signaling Pathways Regulate UV-B Tolerance in Rhododendron chrysanthum Leaves"

_genes, 2023, doi:10.3390/genes14061153_

Round 1

Reviewer 1 Report

Authors investigated the influence of UV-B on transcriptome and phosphoproteome changes in R. chrysanthum. Study is well planned and performed. Article is well written. The material and method section should be better described to assure reproducibility of research.

Following comments should be addressed:

Line 105, 117, 134, 143, 282, 305- Lack of space.

Instead of 3 write three.

Line 108 and 109 write fifteen and three, not numbers.

Section 2.1

Line 100; provide the intensity of UV-B radiation in μmol (photons) m-2 s -1 .

Section 2.3

Provide how the quality of RNA was assured.

Present the volume and concentration of cDNA libraries.

Provide name of reference genome, parameters of raw sequences reads- length, quality necessary to be included into the study.

Software used to analyse raw and clean reads.

How the frequency of reads was measured- FPKM or other measure.  

Section 2.4

Explain if phosphatase inhibitors were addend to stabilize the phosphopeptides.

Line 215- remove „不清” signs

Section 3.2

Provide names of differently phosphorylated proteins in the separate, suplement file or table.

Discussion section

Authors could explain, based on available references, how the increased phosphorylation of ABF2 and SnRK2 could affect their structure and interacions with other proteins important for signal transduction.

Minor editing of English language required.

Author Response

请参阅附件。

Reviewer 2 Report

The article corresponding to the ref. # genes-2410791, entitled “Integration of Phosphoproteomics
and Transcriptome Studies Reveals ABA
signaling pathways regulate UV-B Tolerance in Rhododendron chrysanthum Leaves
”, provides a
characterization of transcriptomic and protein content in leaves under UV exposure conditions. The
issue covered by the article is relevant for the field of research and the objectives of the article. .
English should be edited throughout the text. However, before its acceptance for publication in Genes,
some issues need to be addressed.

Abstract

The first phrase seems redundant (lines 10-11). Please rephrase.

Line 15: factors is no necessary

Line 20: please remove the dot after “that”

Line 22: Remove “In validation experiment” and start the phrase with Plants directly.

Introduction

Line 32: add “This plant” after the dot

Line 37: please remove “and causes a cascade of damage”, reduntat

Line 49: add stomata closure.

Line 50: Please rephrase like The increase of ABA

Line 56: “organ response” not “organ to respond”

Materials and methods

Line 92: please remove “in the same way as” and add are reported

Line 97: please remove “in the verification experiment”

Line 112: remove “Our previous study prepared the transcriptome analysis and”

Line114: remove the f before organism”

Lines 125-126: in the first phrase you “discard the supernatant”. In the follow sentence the
supernatant “was washed”. This supernatant is discared or washed?

Line 126: Which % of acetone?

Line 127: please add the centrifuge condition

Line: 159: “the” in lower case

Results
Lines 207-210: please format following the paper rules

Line 215: please remove the symbol

Line 252: In the figure 5 is reported the model diagram of stomatal closing mechanism not the
expression level of PYR/PYL. Please modify.

No comments
